# Improving Compositionality of Neural Networks by Decoding Representations to Inputs

**Mike Wu, Noah Goodman, Stefano Ermon**
Department of Computer Science
Stanford University
Stanford, CA 94303
{wumike,ngoodman,ermon}@stanford.edu

## Abstract

In traditional software programs, it is easy to trace program logic from variables back to input, apply assertion statements to block erroneous behavior, and compose programs together. Although deep learning programs have demonstrated strong performance on novel applications, they sacrifice many of the functionalities of traditional software programs. With this as motivation, we take a modest first step towards improving deep learning programs by jointly training a generative model to constrain neural network activations to "decode" back to inputs. We call this design a Decodable Neural Network, or DecNN. Doing so enables a form of compositionality in neural networks, where one can recursively compose DecNN with itself to create an ensemble-like model with uncertainty. In our experiments, we demonstrate applications of this uncertainty to out-of-distribution detection, adversarial example detection, and calibration — while matching standard neural networks in accuracy. We further explore this compositionality by combining DecNN with pretrained models, where we show promising results that neural networks can be regularized from using protected features.

## 1 Introduction

Traditional hand-written computer programs are comprised of a computational graph of typed variables with associated semantic meaning. This structure enables practitioners to interact with programs in powerful ways (even if they are not the author) — such as debug code by tracing variables back to inputs, apply assertions to block errors, and compose programs together for more complex functionality. However, traditional software has its limitations: it is difficult to hand-write programs to classify images or extract sentiment from natural language. For these functionalities, deep learning and neural networks [41] have become the dominant approach [23, 2, 46].

While neural networks have made impressive progress on complex tasks, they come at a sacrifice of many of the desirable properties of traditional software. Specifically, the closest approximation to a "variable" in a neural network is an activation. Yet it is difficult to understand a neural network's computation from an activation value, and there is little to associate an activation with semantic meaning. A practitioner cannot write assertion statements to constrain valid neural network logic: checking the values that an activation takes is usually not enough to gauge correctness nor meaning. Moreover, given multiple neural networks, composing them together requires retraining from scratch.

In this paper, we take a modest first step towards bridging the expressivity of deep learning with the engineering practicality of traditional software. We aim to uncover new ways for practitioners to build and use neural networks by leveraging *compositionality*. Specifically, we propose to train a neural network classifier jointly with a generative model whose role is to map the classifier's activations back to the input, approximating invertibility. For any input, a neural network's computation can

35th Conference on Neural Information Processing Systems (NeurIPS 2021).

be represented by a sequence of activations (from each layer), each of which can now be mapped back to the input space. It is this insight that enables a special form of compositionality for neural networks, as inputs derived from activations can be fed back into other models.

In our experiments, we study this compositionality by (1) recursively composing a neural network with itself, creating an ensemble-like model with a measure of uncertainty that is useful for out-of-distribution detection, adversarial example detection, and model calibration; (2) composing neural networks with pretrained models as forms of regularization and distillation; and (3) using decodable activations to discourage a neural network from using protected attributes. Throughout, we show that decodability comes at low cost as we find equivalent accuracy to a standard neural network.

## 2   Background

**Neural Networks**   We will focus on supervised neural networks for classification, although it is simple to extend to regression tasks. We denote a neural network by $f_\theta$ where $\theta$ represents its trainable parameters. A neural network $f_\theta$ maps an example $x$ to predicted probabilities over $K$ classes. Typically, a neural network is composed of $L$ blocks. For example, if $f_\theta$ is a multi-layer perception, each block is a linear layer followed by a non-linearity. If $f_\theta$ is a ResNet [17], each layer is a residual block. In computing its prediction, a neural network $f_\theta$ produces activations $\{h_1, \ldots, h_L\}$ where $h_l$ is the output of the $l$-th block.

To train a neural network, we solve an optimization function of the form

$$\mathcal{L}_{\texttt{nn}}(x; \theta) = \log p(y|f_\theta(x)) + \beta\Omega(x; \theta) \tag{1}$$

using stochastic gradient descent. In Equation 1, the notation $x$ denotes an input example and $y$ its label. The function $\Omega$ represents an auxiliary objective such as a regularizer. The hyperparameter $\beta > 0$ is used to scale the auxiliary loss. For classification, $\log p(y|\cdot)$ is cross entropy.

**Generative Models**   Broadly, we are given a latent variable $H$ and an observed variable $X$. A generative model $g_\phi$ captures the distribution $p(x|h)$, mapping samples of the latent code $h$ to samples

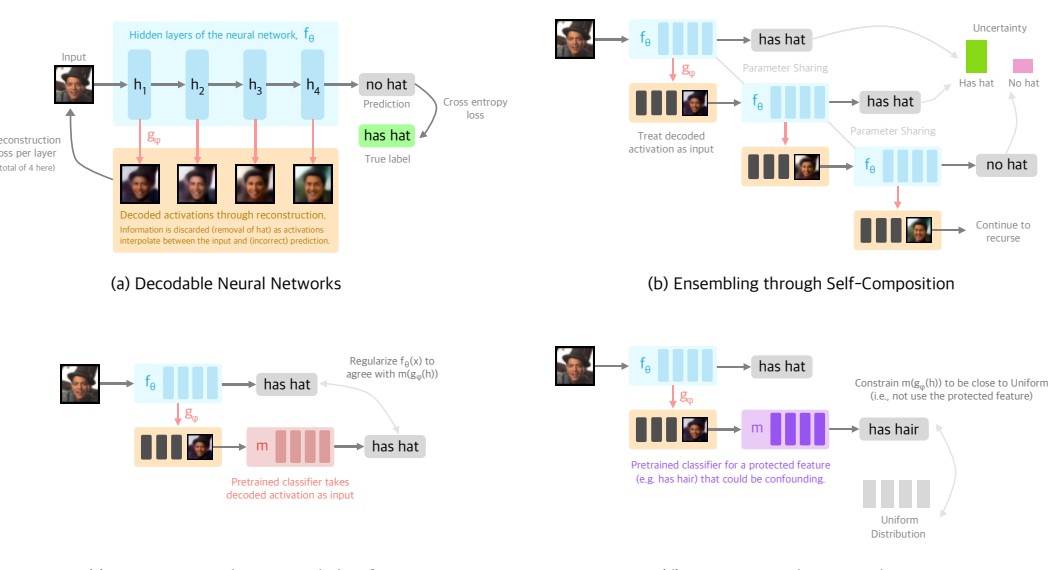

(a) Decodable Neural Networks

(b) Ensembling through Self-Composition

(c) Composition with Pretrained Classifiers

(d) Constraints with Protected Features

Figure 1: *Decodable Neural Networks and Applications*: The decodable neural network (DecNN) is composed of a main network, $f$ and a generative model, $g$. Each activation in $f$ is decoded back to inputs through $g$. Subfigure (a) shows an incorrect classification: the network predicted the man is not wearing a hat despite the true label. Consequently, we see the decoded activations "interpolate" as the hat is removed. Subfigures (b-d) show the compositionality of DecNNs. First, (b) shows self-composition, where a decoded activation can be fed as input back into the network $f$ to create an ensemble with uncertainty. Second, (c) shows how a pretrained classifier, $m$ can be composed with DecNNs for regularization. Finally, (d) shows how to constrain DecNNs to ignore "protected features" through pretrained networks by regularizing for uniformity.

of the observed variable $x$. For a deep generative model, $g_\phi$ is parameterized by a neural network. In the simplest case where $g_\phi$ is an autoencoder, we are also given an encoder $q$ that maps observed samples $x$ to latent codes $h$. Then, we optimize the objective

$$\mathcal{L}_{\text{ae}}(x; \phi) = \log p(x|g_\phi(q(x))) \qquad (2)$$

where for continuous $x$, the expression $\log p(x|h)$ is the log Gaussian probability with mean $h$ and fixed variance (or equivalently, mean squared error). More complex generative models might impose a prior distribution on $h$ [22], an adversarial discriminator [29, 14], or invertibility of $g_\phi$ [35].

## 3    Neural Networks with Decodable Activations

In Section 1, we motivated our desiderata for activations that can be mapped back to the input space. We now propose to do this by "inverting" a neural network at a particular activation. Concretely, given an input $x$, a neural network $f_\theta$, we can compute an activation $h$ by computing the forward pass $f_\theta(x)$. Then, suppose we could construct the set:

$$S(h) = \{\hat{x} : \hat{x} \sim p_\mathcal{D}, f_\theta(\hat{x}) = h\} \qquad (3)$$

where $p_\mathcal{D}$ represents the empirical distribution that $x$ is sampled from. Now, any input $\hat{x} \in S(h)$ is a valid mapping of $h$ back to the input space. (Trivially, $x \in S(h)$.) We call this $\hat{x}$ a *decoding* of the activation $h$. Unfortunately, in practice we do not know how to construct such a set $S(h)$.

But, we do know how to (approximately) sample from $S(h)$. We can train a generative model $g_\phi$ to reconstruct the original input $x$ from neural network activations $h$ (recall $x \in S(h)$). Then, we can sample $x \sim g_\phi(h)$, which we know how to do, to sample from $S(h)$. If we jointly train this generative model $g_\phi$ with the neural network $f_\theta$, we can optimize for activations that both solve the classification task and decode back to inputs at once.

So far, we have only considered a single activation despite our neural network having $L$ such activations. Although one could train a separate generative model per layer, we found it efficient and performant to share weights $\phi$ across all layers. Our joint objective is:

$$\mathcal{L}(x; \theta, \phi) = \log p(y|f_\theta(x)) + \beta \frac{1}{L} \sum_{l=1}^{L} \log p(x|g_\phi(h_l)), \qquad h_1, ..., h_L \text{ come from } f_\theta(x) \qquad (4)$$

where the first term is cross entropy as in Equation 1 and the second term is a reconstruction error (Equation 2) averaged over layers. Comparing Equation 4 to Equation 1, we can interpret the generative loss as a regularization term $\Omega$. Intuitively, in a standard neural network, the activations $h_1, \ldots, h_L$ are unconstrained, chosen only to minimize the loss. However, in Equation 4, the network $f_\theta$ is constrained to produce activations that the generative model $g_\phi$ can invert. In practice, $g_\phi$ is a ResNet-18 decoder with upsampling layers (see Appendix A.1). We call this setup a *Decodable Neural Network*, or DecNN. Figure 1 depicts an illustration of the DecNN architecture.

We highlight that although Equation 4 requires good reconstruction, it does not enforce that reconstructed inputs $f(g_\phi(h))$ must map to the same label $y$ as $f(x)$. We show a few examples of this by visualizing decoded activations throughout a network in Appendix A.3.

## 4    Composing Decodable Neural Networks

In traditional software, we can easily compose two compatible functions $f_1$ and $f_2$ together by $f_2(f_1(x))$ for some input $x$. With this as loose inspiration, we wish to similarly compose neural networks together. As motivation, we may want to do this to faciliate interaction between networks, such as for regularization or distillation. But given two image classifiers, it is not clear *how to tie outputs from one neural network to inputs for the other?* Both networks expect images as input.

The workaround comes from decodable activations. Since they are images of the same shape as inputs, we are free to treat them as inputs for another model. So, we can compose neural networks together using decoded activations as an intermediary: given two models $f_1$ and $f_2$, and an input $x$, we can take an activation $h$ coming from $f_1(x)$, decode it, then provide it to the other model, $f_2(g_\phi(h))$. In the following subsections, we explore two instances of this compositionality: one by composing a model with itself (i.e., $f_1 = f_2$), and the other by composing a model $f_1$ with a pretrained one, $f_2$. See Figure 1b,c for a graphical depiction of both compositions.

## 4.1 Recursive Self-Composition

Our first composition will be to compose a DecNN model with itself. But if we can do that, we can also recursively compose a DecNN model with itself infinitely:

Given a decoded activation $g_\phi(h_l^{(0)})$ for any layer $l$, we can feed it as input back into the classifier, $f_\theta(g_\phi(h_l^{(0)}))$ and maximize agreement with the true label $y$. Intuitively, if $g_\phi(h_l)$ is a good reconstruction of the input $x$ (i.e. $g_\phi(h_l)$ and $x$ belong to the same set $S(h_l)$), then a good classifier $f_\theta$ should predict the correct label with it as input. Now, we also observe that the computation $f_\theta(g_\phi(h_l^{(0)}))$ produces $L$ additional activations, $h_1^{(1)}, \ldots, h_L^{(1)}$ where we use the superscript to represent the number of compositions. We can repeat this exact process with the same intuition, decoding $h_l^{(1)}$ and passing it to $f_\theta$ to produce $h_l^{(2)}$. Computational limits aside, this can be repeated ad infinitum.

One interpretation of this procedure is as data augmentation: we saw in Section A.3 that decoded activations share features relevant to the prediction task (e.g. generated celebrities wear hats) but vary other irrelevant features (e.g. face shape, hair color, etc.). Notably, these generated images are not in the original dataset. Thus, training a model using decoded activations as inputs is similar to augmenting the initial dataset. The only difference is that gradients are propagated back through recursion to the inputs, which is also a form of regularization.

Admittedly, what we have described so far is intractable. Besides recursing infinitely, each activation produces $L$ more activations, creating a tree (with root node $x$) with a large branching factor $L$. In practice, we make two adjustments — First, we limit the recursion to a max depth $D$. Second, instead of constructing the full graph of activations, which would have $O(L^D)$ nodes, we randomly sample "paths" in the graph. Specifically, starting at the root, we compute $\{h_1^{(0)}, \ldots, h_L^{(0)}\}$ through $f_\theta(x)$ and randomly choose an integer $l_0 \in [1, L]$. Then, we compute $\{h_1^{(1)}, \ldots, h_L^{(1)}\}$ through $f_\theta(g_\phi(h_{l_0}^{(0)}))$ i.e. we classify the $l_0$-th decoded activation from depth 0. Again, we randomly pick $l_1$ to compute $\{h_1^{(2)}, \ldots, h_L^{(2)}\}$ through $f_\theta(g_\phi(h_{l_1}^{(1)}))$ and repeat until depth $D$.

Building on Equation 4, we can write this "recursive" objective as:

$$\hat{\mathcal{L}}(x; \theta, \phi) = \log p(y|f_\theta(x)) + \sum_{d=1}^{D} \alpha^d \log p(y|f_\theta(g_\phi(h_{l_{d-1}}^{(d-1)}))) + \beta \sum_{d=1}^{D} \alpha^d \log p(x|g_\phi(h_{l_{d-1}}^{(d-1)})) \quad (5)$$

where the sequence of integers $l_0, \ldots l_{D-1}$ are each sampled uniformly from $[1, L]$. We introduce a new hyperparameter $\alpha \in [0, 1]$ that geometrically downweights later depths (if $\alpha = 0$, Equation 5 reduces to maximum likelihood). The first term in Equation 5 is the usual classification objective. The second term is a sum of cross entropy terms that encourage the decoded activation at depth $d$ to predict the correct label through $f_\theta$. The third and final term is a sum of reconstruction losses that encourage the decoded activation at depth $d$ to approximate the original input $x$.

We call this model a "Recursively Decodable Neural Network", abbreviated ReDecNN. Note that ReDecNN is a special case of an ensemble network. That is, every path sampled in the activation tree by picking $l_0, \ldots l_{D-1}$ builds a legitimate classifier. There are $O(L^D)$ such classifiers implicit in the ReDecNN. We can interpret optimizing Equation 5 as training individual classifiers that are Monte Carlo sampled from the tree every gradient step. See Figure 1b for an illustration.

Our goal in the next few paragraphs is to utilize this ensemble to measure uncertainty.

| Method | MNIST | Fashion | CelebA |
|---|---|---|---|
| Standard | 97.1 (0.11) | 87.7 (0.19) | 90.8 (<0.1) |
| DecNN | **97.8** (0.17) | **88.8** (0.19) | 90.8 (<0.1) |
| ReDecNN | 97.5 (0.20) | 88.1 (0.16) | 90.8 (0.16) |
| Dropout | 95.9 (0.13) | 80.7 (0.24) | 88.5 (0.12) |
| MC-Dropout | 91.3 (0.14) | 78.6 (0.23) | 87.8 (0.19) |
| BayesNN | 96.3 (0.51) | 87.0 (0.36) | 88.2 (0.22) |
| Ensemble | 97.6 (0.11) | 88.4 (0.12) | **90.9** (0.11) |

(a) Classification performance on test set.

| Method | MNIST | Fashion | CelebA |
|---|---|---|---|
| ReDecNN | 0.869 | 0.795 | **0.692** |
| MC-Dropout | **0.878** | **0.818** | 0.657 |
| BayesNN | 0.537 | 0.574 | 0.587 |
| Ensemble | 0.531 | 0.559 | 0.605 |

(b) ROC-AUC of separating correctly classified and misclassified examples using Eq. 6 as a score. The stdev. for all entries over 3 runs are <0.01.

**Measuring Uncertainty**  As a first step, we compare ReDecNN to several baselines: a standard neural network, a neural network with dropout, two kinds of Bayesian neural networks — Monte

Carlo dropout [11], abbreviated MC-Dropout, and weight uncertainty [4], abbreviated BayesNN —
and a naive ensemble network where we train $D$ copies of the standard neural network with different
initializations. For all experiments, see Appendix A.1 and A.2 for training and task details. Table 1a
shows accuracies over a held-out test set, averaged over three runs. While we observe lower accuracy
from Bayesian NNs, we observe (perhaps surprisingly) equal performance of DecNN and ReDecNN
to a standard neural network. Typically, adding uncertainty to neural networks comes at a cost to
performance (as is true for other baselines), but this seems to not be the case here.

The next step is to compare the quality of model uncertainty. For the ReDecNN, for a given input $x$,
we measure uncertainty as follows: sample $N$ different classifiers from the activation tree, and make
predictions with each classifier on $x$. The uncertainty is the entropy of the predictions:

$$\text{Uncertainty}(x) = -\sum_{c=1}^{K} \hat{p}(y = c) \log \hat{p}(y = c) \tag{6}$$

where $\hat{p}(y = c)$ is the empirical probability (e.g. normalized count) of predicting class $c$ out of $N$
classifiers. A higher uncertainty metric would represent more disagreement between classifiers. We
can compute the same metric for a naive ensemble, as well as for MC-Dropout (where repeated calls
drop different nodes) and BayesNN (by sampling weights from the learned posterior).

We want to test if the model's uncertainty is *useful*. One way is to correlate the uncertainty with
when the model makes prediction errors (we call this "misclassification"). It would be useful if the
model was less uncertain on correct predictions and more uncertain on incorrect ones. A practitioner
could then use uncertainty to anticipate when a model might make a mistake. Table 1b reports the
area under the ROC curve, or ROC-AUC of separating correctly and incorrectly classified examples
in the test set (using uncertainty as a score). A higher ROC-AUC, closer to 1, represents a better
separation. We find that ReDecNN is competitive with MC-Dropout, even out performing it on
CelebA. In Section A.5, we include examples of images with low and high uncertainty.

**Out-of-Distribution Detection** A second application of uncertainty is detecting out-of-distribution
(OOD) inputs. Given an input that is far from the training distribution, a model is likely to make
a mistake. Instead, the model should recognize its uncertainty, and refuse to make a prediction.
There is a rich literature on detecting OOD inputs using downstream computation on a trained model
[18, 25, 24, 52, 19]. Unlike those works, we study OOD detection using uncertainty alone.

| | MNIST | | | FashionMNIST | | | CelebA | | |
|---|---|---|---|---|---|---|---|---|---|
| | ReDecNN | MC | Ensemble | ReDecNN | MC | Ensemble | ReDecNN | MC | Ensemble |
| Adversarial (FGSM) | 0.787 | **0.817** | 0.540 | 0.711 | **0.760** | 0.589 | - | - | - |
| OOD (MNIST) | - | - | - | 0.793 | **0.864** | 0.501 | **0.647** | 0.548 | 0.591 |
| OOD (FashionMNIST) | 0.812 | **0.888** | 0.534 | - | - | - | **0.641** | 0.572 | 0.599 |
| OOD (CelebA) | 0.893 | **0.943** | 0.675 | 0.753 | **0.793** | 0.704 | - | - | - |
| Corrupt (Mean) | 0.727 | **0.785** | 0.566 | 0.676 | **0.691** | 0.606 | **0.686** | 0.569 | 0.632 |
| Corrupt (Stdev) | 0.075 | 0.113 | 0.087 | 0.072 | 0.108 | 0.106 | 0.038 | 0.017 | 0.016 |

Table 2: ROC-AUC of predicting which examples are out-of-distribution (OOD). We vary OOD examples to be
adversarial, corrupted, or from a different dataset. Standard deviation for all entries are < 0.01 over three runs.

We explore three kinds of OOD examples: (1) adversarial examples crafted for a single classifier
using FGSM [15], (2) examples from a different dataset (e.g. train on MNIST and use FashionMNIST
as OOD) as done in [25], and (3) corrupted examples by 14 image transformations (e.g. adding pixel
noise), borrowed from [31]. See Appendix A.4 for expanded corruption results.

Table 2 reports the ROC-AUC of separating inlier examples taken from the test split of the dataset the
models were trained on, and outlier examples. From Table 2, we observe ReDecNN is just under
MC-Dropout, but while MC-Dropout achieves these results at the cost of classification accuracy,
ReDecNN does not (Table 1a). Furthermore, we find ReDecNN generalizes better to CelebA, a more
complex image dataset, where it outperforms MC-Dropout (and other baselines).

Focusing on CelebA, we study a domain-specific OOD challenge by holding out all images with
positive annotations for a single attribute as out-of-distribution. Here, inlier and outlier examples are
very similar in distribution. We report ROC-AUCs in Table 3 for three held-out attributes (randomly
chosen): is the celebrity in the image "wearing a hat", has "blonde hair", or "bald"? As this is a more
challenging problem (since inlier and outlier examples are from the same dataset versus different
datasets), the performance is lower than in Table 2. But while MC-Dropout, BayesNN, and a naive
ensemble all perform near chance, ReDecNN is consistently near 0.6.

| OOD: Wearing a Hat | | OOD: Blond Hair | | OOD: Bald | |
|---|---|---|---|---|---|
| Method | ROC-AUC | Method | ROC-AUC | Method | ROC-AUC |
| ReDecDNN | **0.604** (<0.01) | ReDecDNN | **0.593** (<0.01) | ReDecDNN | **0.615** (<0.01) |
| MC-Dropout | 0.519 (<0.01) | MC-Dropout | 0.502 (<0.01) | MC-Dropout | 0.502 (<0.01) |
| BayesNN | 0.510 (0.01) | BayesNN | 0.508 (0.01) | BayesNN | 0.508 (<0.01) |
| Ensemble | 0.526 (<0.01) | Ensemble | 0.509 (<0.01) | Ensemble | 0.503 (<0.01) |

Table 3: We hold out a group of CelebA attributes, such as those wearing a hat, when training. We compute the ROC-AUC of labeling the held-out group as OOD (mean and stdev. over 3 runs).

**Calibration**   A third application of uncertainty we explore is calibration of predicted probabilities. Standard neural networks are notoriously over-confident and are incented in training to predict with extreme probabilities (e.g. 0.99). A model with useful uncertainty would make predictions with proper probabilities. Although many calibration algorithms exist [33, 16], we want to measure how calibrated each model is out-of-the-box to compare the quality of model uncertainties.

Table 4a reports the expected calibration error, or ECE [32], which approximates the difference in expectation between confidence and accuracy using discrete bins. A lower number (closer to 0) represents a more calibrated model. We observe that while a standard neural network has a relatively high ECE, many of the approaches (dropout, MC-Dropout, BayesNN, and ReDecNN) reduce ECE. On the other hand, naive ensembles increase the calibration error, as they are more prone to overfitting. In two of three datasets, ReDecNN achieves the lowest ECE whereas MC-Dropout achieves the lowest in the third. These results should be viewed in parallel to Table 1a, which measures "sharpness". Otherwise, we can trivially reduce ECE to zero by ignoring the input.

| Method | MNIST | Fashion | CelebA |
|---|---|---|---|
| Standard | 1.18 (0.10) | 0.56 (0.14) | 2.75 (0.13) |
| ReDecNN | **0.33** (0.05) | 0.21 (0.03) | **1.92** (0.13) |
| Dropout | 0.63 (0.09) | 0.48 (0.09) | 2.51 (0.13) |
| MC-Dropout | 0.41 (0.06) | **0.13** (0.03) | 2.99 (0.15) |
| BayesNN | 0.91 (0.20) | 0.18 (0.03) | 2.41 (0.19) |
| Ensemble | 1.80 (0.12) | 0.89 (0.07) | 3.12 (0.12) |

| Depth | Acc. | OOD (Fashion) | OOD (CelebA) |
|---|---|---|---|
| 2 | **97.7** | 0.665 | 0.752 |
| 4 | 97.6 | 0.788 | 0.804 |
| 6 | **97.7** | 0.805 | 0.868 |
| 8 | 97.5 | **0.812** | **0.893** |

(a) We compare the expected calibration error (ECE) of ReDecNN to a standard neural network with various regularization and ensembling.

(b) Effect of recursion depth $D$ on classification accuracy and OOD detection for a ReDecNN model (8-layer MLP) trained on MNIST.

**Effect of Depth**   Finally, we study the effect of depth on the efficacy of ReDecNN. For the experiments above, we chose the recursive depth to be equal to the number of layers in the MLP ($D = L = 8$). In Table 4b, we vary the depth from 2 to 8 (while keeping the number of layers fixed to 8) and study the effect on classification accuracy and OOD detection. We find increasing OOD performance as $D$ increases, although with marginally decreasing gains. Moreover, we observe constant accuracy, matching standard neural networks regardless of the choice of $D$.

### 4.2 Composing with Pretrained Models

Apart from self-composition, we can also compose our neural networks with off-the-shelf pretrained models as a form of regularization or distillation. Suppose we have a pretrained model $m$ that maps an input $x$ to a class prediction. We can modify the ReDecNN objective as follows:

$$\tilde{\mathcal{L}}(x;\theta,\phi) = \hat{\mathcal{L}}(x;\theta,\phi) + \sum_{d=1}^{D} \alpha^d \log p(f_\theta(g_\phi(h_{l_{d-1}}^{(d-1)}))|m(g_\phi(h_{l_{d-1}}^{(d-1)}))) \tag{7}$$

where $\hat{\mathcal{L}}$ is as defined in Equation 5. The additional term acts as a divergence, bringing the classifer's predictions close to those of the pretrained model. A similar edit can be made to the DecNN. We revisit Section 4 experiments now using two pretrained models — a linear classifier or a ResNet-18 classifier. For each, we optimize Equation 7 and compute performance on OOD detection.

Table 5 compares the results of composing with either (1) a linear model, (2) a residual network, or (3) no composition (None). Using a linear model, we observe a slight drop in test accuracy but an increase across all OOD experiments and calibration. In fact, these new results rival or surpass MC-Dropout from Table 2. Conversely, with ResNet-18, we observe a 1 point increase in test

| Method | MNIST | | | CelebA | | |
| | None | Linear | ResNet | None | Linear | ResNet |
|---|---|---|---|---|---|---|
| Accuracy | 97.50 (0.20) | 97.17 (0.12) | **98.30** (0.24) | 90.80 (<0.1) | 89.76 (0.11) | **91.28** (0.13) |
| Misclassfication | 0.869 (<0.01) | **0.904** (<0.01) | 0.742 (<0.01) | 0.692 (<0.01) | **0.732** (<0.01) | 0.615 (<0.01) |
| Adversarial | 0.787 (<0.01) | **0.805** (0.01) | 0.729 (<0.01) | - | - | - |
| OOD (MNIST) | - | - | - | 0.647 (<0.01) | **0.693** (<0.01) | 0.609 (<0.01) |
| OOD (Fashion) | 0.812 (<0.01) | **0.845** (<0.01) | 0.650 (<0.01) | 0.641 (<0.01) | **0.673** (<0.01) | 0.585 (<0.01) |
| OOD (CelebA) | 0.893 (<0.01) | **0.928** (<0.01) | 0.862 (<0.01) | - | - | - |
| Corruption (Mean) | 0.727 (<0.01) | **0.790** (<0.01) | 0.694 (<0.01) | 0.686 (<0.01) | **0.708** (<0.01) | 0.616 (<0.01) |
| Calibration (ECE) | 0.334 (0.05) | **0.296** (0.01) | 0.385 (0.02) | **1.927** (0.13) | 1.937 (0.04) | 3.002 (0.10) |

Table 5: OOD detection using uncertainty for ReDecNN composed with pretrained models.

accuracy but notable drops in OOD and calibration results. These polarizing outcomes show two use cases of pretrained models: using simpler models allow for regularization that trades off accuracy for robustness while more complex models encourage distillation, making the opposing tradeoff.

| Method | Eq. 7 | Direct Reg. |
|---|---|---|
| Accuracy | 97.17 | 96.72 |
| Misclass. | **0.904** | 0.828 |
| Adversarial | **0.805** | 0.707 |
| OOD (Fashion) | **0.845** | 0.699 |
| OOD (CelebA) | **0.928** | 0.782 |
| Corruption (Mean) | **0.790** | 0.667 |
| Calibration (ECE) | 0.296 | **0.280** |

Table 6: Composition (Eq. 7) versus direct regularization with pretrained models on MNIST.

An open question is how much Equation 7 benefits from regularizing $f_\theta(g_\phi(h))$ to be close to $m(g_\phi(h))$ versus regularizing it to be close to $m(x)$. The former is a pretrained embedding of a reconstruction whereas the latter is of a static input. Table 6 compares the two on MNIST, where we call the latter "direct regularization". We observe that Equation 7 outperforms direct regularization (other than calibration), often by a large margin. We reason this to be because the input $x$ and the reconstruction $g_\phi(h)$ can be very different, especially in later layers as shown in Appendix A.3. This implies that $m(x)$ and $m(g(h))$ can be very different as well. Intuitively, constraining the model to be similar to what a pretrained model thinks of the reconstruction (which is a dynamic value that varies with model weights) is a stronger learning signal than what the pretrained model thinks of the original input (which is static). In other words, regularizing $f_\theta(g_\phi(h_k))$ to be like $m(x)$ at all layers $l$ is too strict of a constraint.

## 5 Constraining Decodable Computation

A milestone for deep learning would be the ability to specify what information an activation should *not* capture. This is especially important in cases when we have protected attributes that should not be abused in optimizing an objective. While this milestone remains out of grasp, we take a small step towards it by optimizing a neural network to "be indifferent" with respect to a protected attribute through composition with pretrained models.

Suppose we have access to a pretrained classifier $m$ that predicts the assignment for a $K$-way protected feature. Then, building on Section 4.2, we can optimize our classifier to "ignore" information about the protected feature through $m$. To do this, we define the objective:

$$\tilde{\mathcal{L}}(x; \theta, \phi) = \hat{\mathcal{L}}(x; \theta, \phi) + \sum_{d=1}^{D} \alpha^d \log p(m(g_\phi(h_{l_{d-1}}^{(d-1)})) | \frac{1}{K}\mathbf{1}_K) \tag{8}$$

where $\mathbf{1}_K$ is a $K$-dimensional vector of ones. That is, Equation 8 encourages the decoded activation $g_\phi(h_{l_{d-1}}^{(d-1)})$, when passed through the protected classifier $m$, to predict chance probabilities i.e. have no information about the protected attribute. A high performing classifier trained in this manner must have solved the prediction task without abusing the protected attribute(s).

To test this, we extract two attributes from CelebA: "beardedness" and "baldness". Suppose we wish to design a neural network to predict beardedness. If we were to naively do so, there could be a discrepancy in performance between groups of individuals — for example, between bald individuals and individuals with hair, the latter group is more common in the CelebA dataset. Indeed, Table 7 shows a 10 point difference in F1 and a 5 point difference in average precision (AP) in classifying beardedness between the two groups. The second row of Table 7 provides a baseline ReDecNN

| Model | F1 (Bald / Not Bald) | AP (Bald / Not Bald) | ECE (Bald / Not Bald) |
|---|---|---|---|
| Standard | $0.338/0.438$ (<0.01 / <0.01) | $19.4/23.9$ (0.08 / 0.09) | $4.76/3.56$ (0.10 / 0.06) |
| ReDecNN (Eq. 4) | $0.327/0.484$ (<0.01 / <0.01) | $18.6/22.3$ (0.11 / 0.14) | $4.74/3.54$ (0.08 / 0.07) |
| ReDecNN (Eq. 8) | $\mathbf{0.461/0.501}$ (0.01 / <0.01) | $\mathbf{27.2/28.9}$ (0.10 / 0.10) | $\mathbf{3.80/3.28}$ (0.12 / 0.09) |

Table 7: We use a pretrained model on CelebA that predicts baldness to optimize the activations of a second classifier to ignore baldness when predicting beardedness.

optimized without knowledge of the protected attribute. Unsurprisingly, we find a similar discrepancy between groups, as with a standard neural network. In the third row, we evaluate a ReDecNN that was optimized to ignore the "baldness" attribute using a pretrained classifier for baldness. Critically, we find much more balanced (and higher) F1, AP, and ECE across the two groups.

| Method | F1 |
|---|---|
| Standard | 0.383 |
| ReDecNN (Eq. 5) | 0.371 |
| ReDecNN (Eq. 8) | **0.108** |

Table 8: Quantitatively accessing ability to classify baldness.

One unsatisfying but plausible explanation for Table 7 is that the generative model learned to construct adversarial examples such that the pretrained model $m$ cannot predict the protected feature, but that the protected feature is still being used to make predictions by $f$. To show this is not the case, we take a trained ReDecNN optimized by Equation 8, freeze its parameters, and fit a small linear head on top of the last hidden layer to predict the protected feature. If the model learned to ignore this feature, it should not be able to perform the task well. Table 8 reports a poor F1 score (0.1) whereas doing the same with a standard neural network or with an unconstrained ReDecNN gets 3 times the F1 score.

# 6 Generalizing to other Modalities

While we have focused on image classification, the proposed approach is more general and can be extended to other modalities. We apply the same decodable representations to speech, in particular utterance and action recognition. We compare DecNN and ReDecNN to the same baselines, measuring accuracy and uncertainty through similar experiments as we did for images.

| Method | Acc | Misclass. | OOD | ECE | Method | Acc | Misclass. | OOD | ECE |
|---|---|---|---|---|---|---|---|---|---|
| Standard | 94.5 (0.1) | - | - | - | Standard | 42.4 (0.4) | - | - | - |
| DecNN | 93.8 (0.2) | - | - | - | DecNN | 41.4 (0.7) | - | - | - |
| ReDecNN | 93.4 (0.2) | 0.766 | 0.705 | **0.225** (0.1) | ReDecNN | 41.2 (0.5) | **0.642** | **0.605** | 0.523 (0.2) |
| MC-Dropout | 82.1 (0.4) | **0.788** | **0.745** | 0.429 (0.2) | MC-Dropout | 34.5 (0.7) | 0.629 | 0.562 | **0.515** (0.2) |
| BayesNN | 91.6 (0.7) | 0.545 | 0.518 | 1.039 (0.4) | BayesNN | 40.3 (1.2) | 0.523 | 0.500 | 0.918 (0.1) |
| Ensemble | **96.3** (0.1) | 0.529 | 0.506 | 1.103 (0.2) | Ensemble | **44.1** (0.1) | 0.541 | 0.522 | 1.189 (0.5) |
| (a) AudioMNIST | | | | | (b) Fluent Speech Commands | | | | |

Table 9: Performance on speech classification. If not specified, stdev. is <0.01 over three test runs.

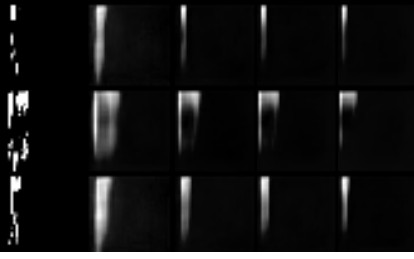

Figure 2: Decoding activations to images for Fluent spectrograms.

We utilize the AudioMNIST [3] and Fluent speech commands [27] datasets, the former being a corpus of 60k utterances of the digits 0 through 9, and the latter a corpus of 100k recordings of 97 speakers interacting with smarthome appliances for one of six actions. Audio waveforms are preprocessed to log Mel spectrograms, outputting a 32 by 32 matrix [47]. Figure 2 shows the input on the leftmost column along with the 2nd, 4th, 6th, and 8th decoded activations for 3 random test examples. Table 9 reports the findings, where like the image experiments, we find ReDecNN has strong performance on OOD detection and calibration in return for only a small drop in performance compared to a standard neural network.

# 7 Related Work

**Autoencoders**    Although DecNNs are reminiscent of autoencoders, there are differences: while autoencoders only reconstruct the final hidden layer, DecNN has a reconstruction for every single hidden layer. This, and because DecNNs have a supervised objective (which autoencoders do not have) critically changes what information is captured in the hidden layers. Whereas the autoencoder seeks perfect reconstruction, the DecNN does not, as evidenced by loss of information for reconstructions later in the network (see Appendix A.3). Finally, we point out that the applications of autoencoders are very different from the DecNN. The former is not used for calibration or composition.

**Invertible and Ensemble Networks**    The DecNN is akin to invertible generative models [6, 36] as DecNN is training a generative model to "invert" the classifier up to a hidden layer. However, unlike invertible flows, DecNN does not impose an explicit prior on the activations (i.e. latent variables), and further, DecNN has a supervised objective. Our proposed recursive network, ReDecNN, has similarities to Bayesian neural networks [11, 4] and ensemble networks [48] — two baselines we compare against for evaluating model uncertainty. However, we find ReDecNN to be easier to train than doing inference over weights, and cheaper in parameter count than naive ensembles.

**Probing Neural Networks**    A related line of work seeks to probe neural network post-training to understand the underlying computation [1]. Most relevant to our work is an approach that inverts CNN activations through reconstruction [28]. Unlike this approach, we are not proposing any additional computation post-hoc. Rather, we are interested in *optimizing* neural networks such that their representations are more easily mapped back to the input domain.

**Robustness and Protected Attributes**    Out-of-distribution detection [26, 24, 52, 40, 20, 19], selective classification [12, 13], adversarial perturbation detection [49, 10, 30, 34], and neural network calibration [16, 51, 50, 32, 38] each have a rich subfield of algorithms. While task-specific methods likely outperform our results, we are excited about representations that enable uncertainty without task information or additional algorithms. Our approach also shares similarities to [42] where the authors minimize the mutual information between learned representations and a protected attribute. We view Equation 8 as an approximation of this where we treat the pretrained classifier for the protected attribute as a proxy for the mutual information.

# 8 Limitations and Future Work

| Method | MNIST | Fashion | CelebA |
|--------|-------|---------|--------|
| Standard | 13.6 (0.3) | 13.5 (0.2) | 82.6 (6.2) |
| DecNN | 80.5 (6.8) | 99.2 (9.6) | 486.2 (12.9) |
| ReDecNN | 96.7 (9.7) | 78.6 (6.6) | 521.6 (9.2) |
| Dropout | 13.9 (0.6) | 14.0 (0.1) | 79.5 (5.7) |
| MC-Dropout | 13.9 (0.6) | 14.0 (0.1) | 79.5 (5.7) |
| Ensemble | 109.6 (7.1) | 132.6 (8.5) | 659.4 (8.2) |

Table 10: Cost (seconds) of 1 epoch on a Titan X GPU (averaged over 10 epochs).

To close, we discuss a few limitations. First, our approach is bottlenecked by the quality of the generative model. Without a good reconstruction, optimization will be intractable. However, in light recent work [45, 44, 5, 21, 43], this is becoming less of a problem as new generative models surface. Second, in the main text, we only explored feedforward classifiers for simplicity. Our approach extends naturally to residual and transformer blocks, and future research could explore this direction. As a start, we provide experiments for extensions to CNN architectures in Appendix A.6. Third, optimizing recursive networks, although parameter efficient, costs more compute as backpropagation is more expensive. See Table 10 for timings of 1 epoch in seconds. The proposed method does have significant computational overhead, averaging about 6x cost, while baseline methods like Dropout and MC-Dropout impose little to no overhead. On the other hand, ReDecNN has comparable cost to DecNN despite requiring recursive gradients. Also, both DecNN and ReDecNN are both cheaper than naive ensembles, significantly in the CelebA case. Future work could explore weight sharing across layers (not just depth) to reduce compute.

In summary, we explored building neural networks with decodable representations, and leveraged this new property to compose models together. By re-purposing decoded activations as novel inputs, we were able to join neural networks together in useful ways for out-of-distribution detection and calibration, as well as influence what information a network retains in optimization. We are optimistic about promising results and look to future research for more applications of decodability.

## Acknowledgments and Disclosure of Funding

We thank the Ermon group for their comments and suggestions. We thank the reviewers for their many iterations of feedback that helped this paper improve significantly. This research was supported in part by NSF (#1651565, #1522054, #1733686), ONR (N000141912145), ONR MURI (N000141612007), AFOSR (FA95501910024), ARO (W911NF-21-1-0125), and Sloan Fellowship. MW is supported by the Stanford Interdisciplinary Graduate Fellowship as the Karr Family Fellow.

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
