# A Appendix

## A.1 Training Details

In our experiments, the classifier $f_\theta$ is a 8-layer MLP with 128 hidden dimensions per layer. We assume the same dimensionality per layer for simplicity but the approach easily supports MLPs with variable hidden dimensionalities: one can train seperate generative models or use the technique from [39] where smaller dimensionalities are padded. The generative model $g_\theta$ is a ResNet18 decoder that maps a 128 dimensional vector to a 3 by 32 by 32 pixel image for RGB images and 1 by 32 by 32 for grayscale images and speech spectrograms. We repurpose the implementation from the PyTorch Lightning Bolts [9] repository: `https://github.com/PyTorchLightning/lightning-bolts/blob/master/pl_bolts/models/autoencoders/components.py` (Apache 2.0 License). (All input images are reshaped to 32 by 32 pixels. No additional image transformations were used in training nor evaluation.) All models are trained for 50 epochs with batch size 128, Adam optimizer with learning rate 1e-4. We set $\beta = 10$ in all cases. For all ReDecNN models, we use a max depth $D = 8$ and set $\alpha = 0.5$ in all cases. To make for a fair comparison, for all naive ensemble networks, we train 8 copies of a neural network with different weight initializations. For baseline models using dropout and MC-Dropout, we use 0.5 dropout probability. For weight uncertainty (i.e. BayesNN), we use the Blitz library [7], `https://github.com/piEsposito/blitz-bayesian-deep-learning` (GNU V3 License). All pretrained models used were ResNet-18 classifiers. Our ResNet classifier implementation was adapted from `torchvision`. For each of the architectures, to support 32x32 images (which are smaller than the standard), we replace the first 7x7 convolutional layer with a 3x3 convolutional layer and reduce the first max pooling layer. We did not use their pretrained weights from `torchvision` and instead trained them ourselves on MNIST, FashionMNIST, and CelebA from scratch (again 50 epochs). All models were trained on a single Titan Xp GPU with 4 CPUs for data loading. An average model takes 1 hours to train for MNIST and FashionMNIST and 4 hours for CelebA, and speech experiments. In our experiments, we compute uncertainty using 30 random samples (e.g. for ReDecNN, this is 30 paths; for MC-Dropout, this is 30 different dropout configurations, etc.). ROC-AUC computation is done through `scikit-learn` (BSD License). In CelebA, only a subset of 18 attributes are used, chosen for visual distinctiveness as done in [37]. Conveniently, this removes many trivial features that are mostly of a single class as well. In experiments, we utilize the PyTorch Lightning framework [8] (Apache 2.0 License) and Weights and Biases (MIT License) for tracking. For speech experiments, the AudioMNIST dataset is found at `https://github.com/soerenab/AudioMNIST` (MIT License) and the Fluent dataset can be downloaded at `https://fluent.ai/fluent-speech-commands-a-dataset-for-spoken-language-understanding-research/` (academic license). All speech spectrograms are normalized using a mean and standard deviation computed from the training dataset. We use `torchaudio` and `librosa` to efficiently compute Mel spectrograms.

## A.2 Dataset & Task Details

We describe the setup for each set of results presented in the main text, paying close attention to the data used, and any special setup required.

**Table 1a** For classification, we use datasets as in standard practice. MNIST and FashionMNIST are both classification tasks with 10 classes. CelebA has 18 binary attributes, and we optimize an objective containing the sum of 18 binary cross entropy terms, one for each attribute. Accuracy is reported using the respective test set.

**Table 1b** Given all test examples, we make predictions and get uncertainty scores using the trained model. We split the test set into two groups, those we correctly classified, and those we misclassified. Then, we compute the ROC-AUC with uncertainty as the prediction score. An ROC-AUC of 1 would mean that the model perfectly assigned high uncertainty to examples it misclassified.

**Table 2** OOD detection experiments are very similar to the misclassification experiments except we define groups differently. Generally, there is inlier group and an outlier group. We compute uncertainty scores over both groups and then compute ROC-AUC with uncertainty as the prediction score. An ROC-AUC of 1 would mean that the model perfectly assigned high uncertainty to outlier examples. Table 2 has three different ways of defining inliers and outliers. First, fixing a dataset and trained classifier, we can compute adversarial examples with FGSM on the test set. This serves as the

outlier set while the original test set serves as the inlier. Second, for rows labeled OOD (XXX), we take the test set from dataset XXX to be the outlier set while the test set from training dataset acts as the inlier set. Finally, for corruptions, we compute image transformations on the test set (e.g. blur all image in the test set) and treat this as the outlier set (again, the original test set acts as the inlier set). We do this separately for each corruption transformation, and report the average and standard deviation (over corruptions) in the main text.

**Table 3**   In Table 2, we limited OOD experiments to using a separate dataset as outliers. For CelebA, we can consider a more difficult task by holding out all images with a positive label for a single attribute. We consider three such attributes: hold out all people wearing a hat, all people with blond hair, or all people who are bald. Because a model with uncertainty has not seen any images of people wearing hats (for example), it should assign these images higher uncertainty. The difficulty of this task comes from outlier and inlier inputs being very similar in features; they are now both images of celebrities, rather than images of digits. In all cases, the inlier set is defined to be all test set images without a positive label for the held out attribute.

**Table 4a**   ECE is computed on predicted probabilities using the test set for MNIST, FashionMNIST, and CelebA seperately. The labels used for binning are the standard dataset annotations.

**Table 4b**   Same setup as in Table 2 but we vary the recursion depth for ReDecNN.

**Table 5**   Design of experiments in the Table are as described above for misclassification, OOD, and calibration. We pretrain a linear model and a ResNet-18 model using the training set of MNIST and CelebA, separately. For CelebA, pretrained models optimize a sum of 18 binary cross entropy terms.

**Table 6**   Same setup as in Table 5.

**Table 7**   Using the CelebA dataset, we discard all attributes except the "baldness" and "beardedness" columns (note this does not change the size of the dataset). We fit a ResNet-18 model to predict baldness only (here, we do not use the beardedness label). Following this, we train a ReDecNN with Equation 8 to ignore the protected attribute (baldness) when predicting beardedness. We then compute the F1, AP, and ECE scores separately for two groups, one containing images of all bald individuals in the test set and one containing images of all non-bald individuals in the test set. These two groups will not be equal in size but are sufficiently large to compute statistics.

**Table 8**   Given the CelebA dataset, we only use the "baldness" attribute. Given a trained model (a standard neural network or a ReDecNN), we freeze its parameters, find the last hidden layer (the $L$-th one), and initialize linear head on top. This linear head returns a binary prediction: bald or not bald. We optimize this model with binary cross entropy and see if the frozen weights contain the information to predict baldness.

**Table 9**   See Appendix A.1 for the dataset links for AudioMNIST and Fluent Speech Commands. We preprocess waveforms in each to be mel spectrograms of fixed size 1x32x32, mimicking the CIFAR10 setup (note that there is only 1 channel). See Figure 2 left-most column for three examples of input spectrograms. AudioMNIST has 10 output labels whereas Fluent Speech Commands has 6 output labels. The task and data setup remain the same as in Table 2 for OOD, misclassification, and calibration experiments.

**Table 10**   Given a dataset and model, we use the `time.time` function in Python to record the start and end times for each epoch on a Titan X GPU. We do so for 10 contiguous epochs and report the average (and standard deviation). Eight data workers are used to load images (increasing this will decrease the cost per epoch).

**Table 11**   This is identical to Table 2 but does not report the average over corruption experiments, but reports each individually.

**Table 12**   This is identical to Table 2 but replaces an MLP with stacked convolutional layers.

### A.3   Visualizing Decodable Activations

The most straightforward application of decodable activations is that we can visualize them. Figure 3 shows randomly chosen examples taken from DecNNs trained on FashionMNIST and CelebA. In each subfigure, the leftmost column is from the dataset while the remaining eight columns are decoded activations from layer 1 to 8 ($f_\theta$ is an MLP with $L = 8$ layers).

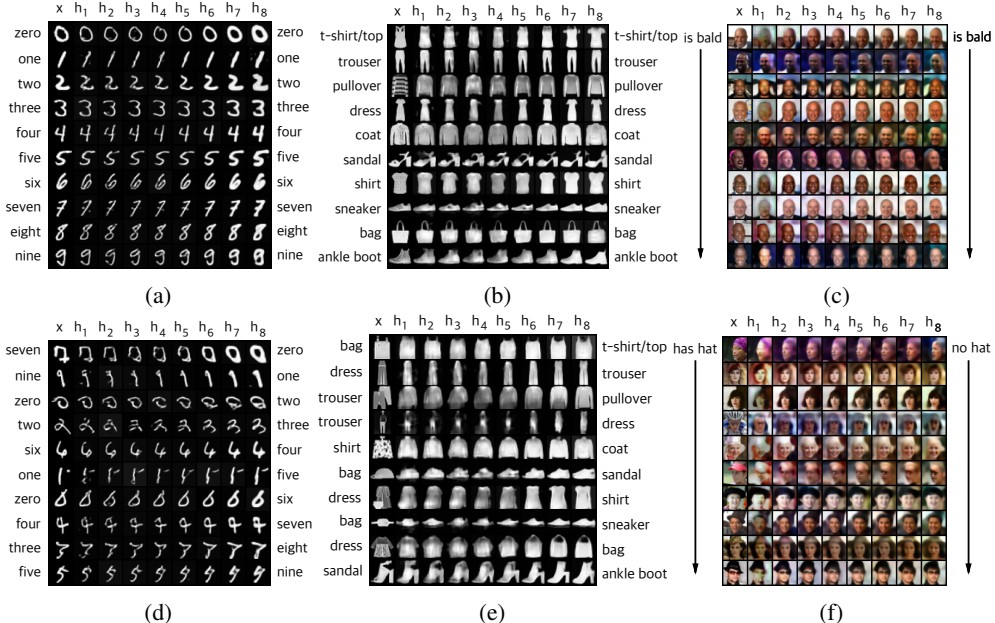

Figure 3: *Visualizing activations*: the top row shows decoded activations when the model correctly classified examples; the bottom row shows misclassifications.

The top row of three subfigures shows decoded activations for correctly-classified examples whereas the bottom row of three subfigures shows mis-classified examples, all randomly chosen. For correctly-classified examples, we observe that the decoded activations tends toward class prototypes. Decoded images from the 7th or 8th activation lose details (e.g. patterns, logos, or shoe straps disappear from clothing). The top row of Figure 3b presents a good example: the tank-top (a rare form of the t-shirt) is projected to a more prototypical t-shirt. Furthermore, we also find that mis-classified examples morph into images of the incorrect class. The seventh row of Figure 3e morphs the two articles of clothing that compose the dress to build a shirt. The seventh row of Figure 3f show the celebrity's hat transform into a background color.

Although we may be tempted to interpret these activations as revealing what the neural network is learning, the objective (Equation 4) does not guarantee this, as reconstructions could appear visually similar to the input $x$ but map to completely different outputs $y$, and as such, serve as a poor explanation as to what the underlying function $f$ is doing. Future work could consider also constraining reconstructions to map to the same label as $f(x)$.

## A.4   Extended OOD Results

We provide a more thorough breakdown of performance for the 14 different image corruptions. In the main paper, we only show the average performance over all corruptions.

## A.5   Visualizing OOD Examples

In the main paper, we present OOD results using ROC-AUC. Here, we visually inspect some examples the model deems as OOD. Figure 4 shows one image from each class (of MNIST, FashionMNIST, and CIFAR10) with low and high uncertainty. We observe that high uncertainty images are less prototypical. For example, the high uncertainty digits in Figure 4d have accented curvature, whereas the high uncertainty clothing in Figure 4e have more atypical designs, and the high uncertainty images of celebrities in Figure 4f, although annotated as not bald, are wearing hats or have thin hairlines and exposed foreheads. On the other hand, low uncertainty images in all datasets look more prototypical.

| | MNIST | | | FashionMNIST | | | CelebA | | |
|---|---|---|---|---|---|---|---|---|---|
| | ReDecNN | MC | Ensemble | ReDecNN | MC | Ensemble | ReDecNN | MC | Ensemble |
| Adversarial (FGSM) | 0.787 | **0.817** | 0.540 | 0.711 | **0.760** | 0.589 | todo | todo | todo |
| OOD (MNIST) | - | - | - | 0.793 | **0.864** | 0.501 | **0.647** | 0.548 | 0.591 |
| OOD (FashionMNIST) | 0.812 | **0.888** | 0.534 | - | - | - | **0.641** | 0.572 | 0.599 |
| OOD (CelebA) | 0.893 | **0.943** | 0.675 | 0.753 | **0.793** | 0.704 | - | - | - |
| Corrupt (Brightness) | 0.851 | 0.955 | 0.702 | 0.732 | 0.667 | 0.816 | 0.653 | 0.558 | 0.623 |
| Corrupt (Dotted Line) | 0.672 | 0.684 | 0.546 | 0.602 | 0.589 | 0.535 | 0.647 | 0.559 | 0.626 |
| Corrupt (Glass Blur) | 0.664 | 0.718 | 0.515 | 0.617 | 0.584 | 0.525 | 0.641 | 0.596 | 0.652 |
| Corrupt (Impulse Noise) | 0.702 | 0.848 | 0.551 | 0.702 | 0.746 | 0.642 | 0.683 | 0.555 | 0.640 |
| Corrupt (Rotate) | 0.689 | 0.714 | 0.523 | 0.754 | 0.839 | 0.600 | 0.645 | 0.585 | 0.652 |
| Corrupt (Shear) | 0.685 | 0.661 | 0.502 | 0.611 | 0.809 | 0.543 | 0.750 | 0.581 | 0.636 |
| Corrupt (Spatter) | 0.647 | 0.683 | 0.502 | 0.617 | 0.615 | 0.534 | 0.715 | 0.569 | 0.641 |
| Corrupt (Translate) | 0.809 | 0.889 | 0.511 | 0.725 | 0.826 | 0.555 | 0.673 | 0.577 | 0.610 |
| Corrupt (Canny Edges) | 0.696 | 0.824 | 0.521 | 0.703 | 0.807 | 0.571 | 0.649 | 0.569 | 0.650 |
| Corrupt (Fog) | 0.868 | 0.940 | 0.732 | 0.748 | 0.670 | 0.810 | 0.729 | 0.608 | 0.643 |
| Corrupt (Scale) | 0.768 | 0.846 | 0.535 | 0.808 | 0.805 | 0.521 | 0.750 | 0.562 | 0.632 |
| Corrupt (Shot Noise) | 0.617 | 0.559 | 0.526 | 0.541 | 0.537 | 0.528 | 0.685 | 0.542 | 0.592 |
| Corrupt (Stripe) | 0.804 | 0.892 | 0.755 | 0.684 | 0.522 | 0.772 | 0.674 | 0.563 | 0.628 |
| Corrupt (Zigzag) | 0.709 | 0.785 | 0.514 | 0.626 | 0.658 | 0.532 | 0.714 | 0.555 | 0.627 |
| Corrupt (Mean) | 0.727 | **0.785** | 0.566 | 0.676 | **0.691** | 0.606 | **0.686** | 0.569 | 0.632 |
| Corrupt (Stdev) | 0.075 | 0.113 | 0.087 | 0.072 | 0.108 | 0.106 | 0.038 | 0.017 | 0.016 |

Table 11: We report the ROC-AUC of predicting which examples are out-of-distribution (OOD) using uncertainty. We vary OOD examples to be adversarial, corrupted, or taken from a different dataset.

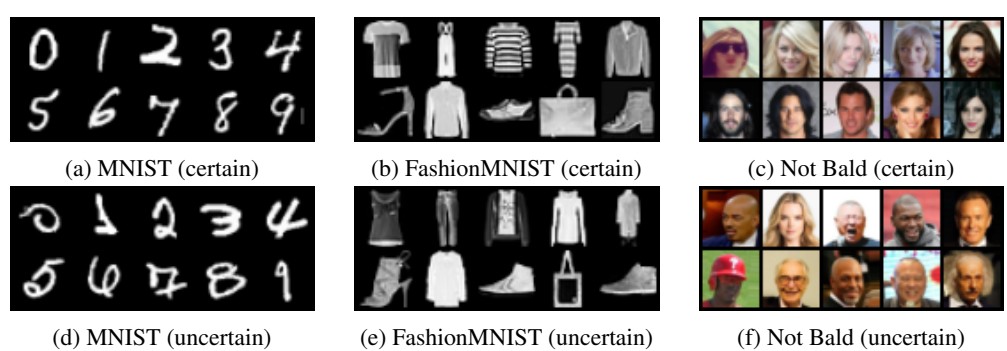

(a) MNIST (certain)    (b) FashionMNIST (certain)    (c) Not Bald (certain)

(d) MNIST (uncertain)    (e) FashionMNIST (uncertain)    (f) Not Bald (uncertain)

Figure 4: *Randomly sampled images with low and high uncertainty.* Images with high uncertainty appear less "prototypical". Figure 4(c,h) shows images from CelebA annotated as "not bald". We observe images that the model is uncertain about depict indviduals with a thin hair line, exposed foreheads, or wearing helmets.

## A.6 Extensions to CNN Architectures

In the main text, we limited the experiments to MLP architectures for simplicity. Here, we include a subset of the experiments in Section 4, replacing the MLP with stacked convolutional layers.

For this experiment, we use architectures of 8 total convolutional layers with ReLU nonlinearity and 64 filters. We use a U-Net decoder (again with 64 filters) as the generative model, which is better suited to convolutional activations (which are now three dimensional rather than two). Like before, we optimize for 50 epochs, batch size 128, learning rate 1e-4, and use Adam. For a baseline, we compare against MC-Dropout, the most competitive baseline from the main text.

| Method | Acc. | Misclass. | OOD (MNIST) | OOD (CelebA) | Corruption (Mean) | Calibration (ECE) |
|---|---|---|---|---|---|---|
| Standard | 90.2 | - | - | - | - | 1.848 |
| DecNN | **90.6** | - | - | - | - | - |
| ReDecNN | 89.5 | **0.876** | **0.883** | 0.844 | 0.712 | **0.837** |
| MC-Dropout | 85.9 | 0.863 | 0.776 | **0.895** | **0.723** | 0.889 |

Table 12: DecNN with convolutional layers trained on FashionMNIST.

From Table 12, we see similar findings to the MLP results, although overall performance is higher due to a more expressive architecture. Namely, ReDecNN, DecNN, and a standard NN have similar test accuracy whereas MC-Dropout has a 4 point lower accuracy. When using the uncertainty score to detect misclassification, OOD, and corruption, we find much closer performance between MC-Dropout and ReDecNN (whereas in the paper, MC-Dropout outperformed ReDecNN on FashionMNIST

consistently), which is promising. Finally, for calibration, MC-Dropout and ReDecNN are again comparable, with the latter having a slightly lower ECE score.