# OpenReview forum: "Improving Compositionality of Neural Networks by Decoding Representations to Inputs"
_NeurIPS.cc/2021/Conference — NeurIPS 2021 Poster_

### Official Review · Reviewer_3B1Z · 2021-07-15

**Rating:** 7
**Confidence:** 4

**Summary:**

This paper explores jointly training generative model probes alongside a neural architecture to imbue the model with capabilities analogous to "debugging" in traditional software. By training the generative model probes to reproduce the model inputs, every intermediate activation can be decoded to input space (images). The authors use this (1) to interpret the partial progress of the model made at intermediate layers, (2) to restrict the model from using sensitive information from the inputs, and (3) as an interface to compose the model with itself and with other pretrained image models. The authors further use self-composition to produce an ensemble from a tree of connected classifiers, which in turn gives uncertainty measurements that enable (4) out-of-distribution detection and (5) calibration. Experimental evidence backs each of these contributions.

**Ethical Concerns:**

The paper considers the task of predicting attractiveness in the CelebA dataset while ignoring skin color, using a pretrained model on CelebA that predicts pale skin. I understand that this is a sensitive topic, and accordingly have selected Yes for sending this paper for an ethics-focused review.

**Ethics Review Area:**

["Inappropriate Potential Applications & Impact  (e.g., human rights concerns)"]

**Limitations And Societal Impact:**

The authors consider both technical limitations and potential societal impact of their work in Section 9.

**Main Review:**

I outline here the main strengths of the paper. The core idea that the paper proposes -- training generative model probes jointly with an image classification -- is novel and conceptually simple, but with powerful implications. This technique allows for several applications, some of which resemble traditional software debugging capabilities (interpreting the partial progress of the model is like a step-through debugger; restricting the model's use of sensitive information is analogous to an assert statement; composing two models or the model with itself using the image domain as the interface is like traditional function composition), and some of which are desirable properties of neural models (out-of-distribution detection, calibrated predictions).

Neural networks usually lack these capabilities. That a single conceptually straightforward addition to neural networks allows for all of these capabilities concurrently is a significant result. That it does this without undue degradation in performance of the original underlying model further improves the significance. The paper does a good job of covering the several benefits the method confers and defending each one with experimental evidence. These benefits span interpretability, fairness, flexibility through model composition, and uncertainty measurement (OOD-example detection and calibration).

This is a comprehensive paper presenting a novel type of optimized neural probe, showcasing the method and several capabilities that it enables in image neural networks. It leverages generative image models as neural probes. Of course neither generative models nor neural network probes for interpretability individually are new, but I am not aware of generative models being used as neural network probes previously. To my knowledge, the use of generative models in this way is original. I recommend investigating the "linear probes" line of work, not discussed in your (otherwise overall quite comprehensive) discussion of the literature. [1]

As with the "observer effect" in physics, the probes that you consider alter the behavior of the underlying network. This is because they are optimized in conjunction with the main model. Fortunately, the paper finds that performance is not degraded by optimizing the probes jointly with the rest of the network. Future work could explore the degree to which the solutions found by the original network change when optimized in conjunction with these generative probes. (Note, unlike with the observer effect, a model trained with the probes can be used for inference with the probes removed, without changing the network's output.)

A key drawback of the approach is that there is a significant increase in compute for training and inference from the addition of the generative model to the underlying model, and accordingly there is a decrease in training speed. The paper suggests otherwise at line 122, and I think it is incorrect to do so; I recommend rewording. The paper would benefit from quantifying this drawback, or at least addressing it in the text.

At line 96 the paper describes sampling from $g_\theta(h)$ as approximating sampling from $S(h)$. I don't think this approximates sampling from $S(h)$, though, since the training data doesn't include any members of $S(h)$ other than $x$. We never once see another input that produces $h$ besides the one that originally produced $h$. Further, I don't think defining $S(h)$ was necessary for your argument.

On the fairness application at line 275, you state that "a high performing classifier trained ... without abusing the protected attribute." Alternatively the model could have constructed adversarial examples such that the pretrained classifier m cannot predict the protected feature, despite the protected feature still being present and still being used to make predictions. An investigation into whether this alternative behavior occurs would benefit the paper or future work. One way to approach this investigation would be to retrain the sensitive attribute classifier to see if the sensitive attribute can still be detected, rather than assuming the pretrained classifier is still a good measure of whether the sensitive attribute is being considered.

The main text of the paper does not mention the details of the generative model. This information is contained in the Appendix. I encourage the authors to include at minimum a reference to this Appendix in the body of the paper.

---

[1] Understanding intermediate layers using linear classifier probes https://arxiv.org/abs/1610.01644

---

Nit at line 71: an non-linearity -> a non-linearity

Nit: activations have same type as encoded training examples, not the training examples themselves.

Typo at line 283: discrepancy is spelled wrong

Nit: No entry is highlighted in OOD column for AudioMNIST.

**Needs Ethics Review:**

Yes

**Time Spent Reviewing:**

8

---

> ### Author Response · Authors · 2021-08-09
> **Author Comments to Reviewer 3B1Z**
>
> We thank the reviewer for their thoughtful comments. We hope to address questions and concerns below:
>
> > Linear probes could be a good addition to the related work of neural probes.
>
> This is a good suggestion! The paper the reviewer mentioned [1] as well as [2] are certainly precursors to our probing work. We will add references in Section 8 for the final draft.
>
> [1] Understanding intermediate layers using linear classifier probes
> [2] SVCCA: Singular Vector Canonical Correlation Analysis for Deep Learning Dynamics and Interpretability
>
> > Future work should look at how decision functions learned by neural networks change when trained with knowledge of the probes.
>
> This is an interesting direction forward (and more generally, I found the connection to the “observer effect” engaging). I agree that optimizing the classifier and the generative model in conjunction changes the underlying loss landscape, and thus the decision functions learned. Further work could try to (1) visualize the loss landscapes through [3] or try to quantify properties of it such as smoothness. More broadly, future work can explore the extent to which our generative probes enables interpretability of neural network behavior.
>
> [3] Visualizing the Loss Landscape of Neural Nets
>
> > The increase in compute for the approach should be documented.
>
> Yes, we agree. Please see the general comment above for runtime comparisons across the different datasets and models we studied.
>
> > The notation with $S(h)$ can be simplified as no other input produces $h$ other than the original $x$ in the training set that produced $h$.
>
> We wanted to present the approach in a general manner. While in our experiments, because $h$ is a high dimensional continuous vector, the reviewer is correct that it is very unlikely that two different inputs $x$ and $x’$ would both map to $h$. Instead, suppose that $f$ is a discrete neural network whose hidden states are quantized to a finite set of possible values. Then, the event that multiple inputs clash to the same hidden representation is much more likely. Alternatively, there may be settings where multiple inputs are optimized to have the same continuous representation (e.g. contrastive learning). The upside of the $S(h)$ notation is its flexibility.
>
> > It is possible that the model could have learned to produce examples that the pre-trained classifier m could not predict the protected feature, rather than truly not using the protected feature.
>
> This is a good question! In our experiments, we treated the classifier as a surrogate “oracle” for the protected feature but it is possible that the model could have learned to fool the classifier. To test this, we adapted the reviewer’s suggestion. After training the model, we can freeze weights of the model and train a small linear head on top to predict the protected feature. If the model learned to ignore this feature, it should not be able to perform the task. (If we do not freeze the weights, the model could trivially discard all prior information and learn to predict the protected feature). We will add the experiment to the supplement.
>
> Method | F1 score on classifying Pale Skin
>
> Standard | 0.32
>
> ReDecNN (Eq. 4) | 0.29  ← Without ignoring protected feature
>
> ReDecNN (Eq. 8) | 0.01  ← With ignoring protected feature
>
> We find that the linear head on a constrained ReDecNN nearly predicts all 0’s (the majority class) while a linear head on a standard NN fares much better.
>
> > Main text should reference generative model architecture, at least.
>
> Good point! In the final draft, we will include an architecture description in Section 3 and reference the supplement for full details.
>
> > Writing comments: typos, missing highlights.
>
> Thank you for pointing these out! We will make the suggested changes.

---

> > ### Comment · Reviewer_3B1Z · 2021-08-24
> > **Thanks for your reply**
> >
> > Thanks for your reply.
> >
> > I want to highlight your response to my concern that the model could produce examples for which the pre-trained classifier might not predict the protected feature, despite the protected feature still being present in the example. Your additional experiment to check for this concern is thoughtful, and the results are promising. Thanks for performing this additional experiment and including it in your supplementary materials.

---

> > > ### Author Response · Authors · 2021-08-27
> > > **Author response to Reviewer 3B1Z**
> > >
> > > Thanks for the suggestion! We will be sure to include this experiment in the final draft supplement.

---

### Official Review · Reviewer_gKMX · 2021-07-16

**Rating:** 8
**Confidence:** 2

**Summary:**

This paper argues for inverting intermediate latent representations of convolutional classifiers -- predicting the original input -- as a tool for improving robustness and interpretability. The ability of the model to learn invertible intermediate hidden representations acts as a regularizer during training.

This basic premise is used in several ways. For example, a recursive version of this idea (using the reconstructed input to classify, and the intermediate hiddens of that to re-reconstruct the input, adding that reconstruction loss into the overall loss calculation) leads to an interesting form of drop out: choose which reconstructions (from which layer, and which recursive depth) to use in the final output. That can be used for enhanced forms of Monte Carlo Dropout, improving uncertainty computations over regular MC Dropout.

Another use is in placing constraints on intermediate hidden layers. For a fairness goal, such a constraint might be that, given a good classifier for some protected attribute, that classifier has poor performance on the reconstructed inputs from the original model's hidden layers. In a sense, this constrains the training of the model to lose the information that would allow the external classifier to predict the protected feature.

**Ethical Concerns:**

No concerns.

**Limitations And Societal Impact:**

No concerns.

**Main Review:**

# Overall Assessment

Originality: Although autoencoders, invertible networks, and interpretability tools are not new, I found the formulation here interesting and elegant. I would have appreciated some statement on the contrast to vanilla autoencoders here. I see some differences, but I'd like to see those in the paper.

Quality: I appreciated the evaluation. It seemed sound. I questioned whether some of the evaluated aspects (e.g., OOD detection) are really a contribution of this work, or a secondary artifact of the improved uncertainty estimation.

Clarity: The write-up is excellent.

Significance: The results seem of academic importance for now, given the lack of a cost-benefit analysis of the approach. However, the idea of getting uncertainty out of the main computation, without further training past the original training seems useful in its own merit, perhaps analogously to Monte Carlo Dropout. I find the novelty also pushes the significance up in my eyes.


# Feedback

I enjoyed reading your work. It's an interesting use of autoencoders that I hadn't seen before. I found the combination of the basic reconstruction-loss term in more intricate formulations fun to read, and a great demonstration of creativity.

Most of my feedback is on the positioning of this work. I found the introduction (and even the title) a bit of an oversell. Obviously, this work isn't solving the problem of neural network interpretability, or enabling arbitrary composition with other neural networks. Your example mentioned "segmentation and classification" which is more about functional compositionality (exciting!), but you only talk in the paper about self-composition, and regularization with a pre-trained model that does the same function. That's why I found that story a bit oversold. Your work does interesting things (using reconstruction from partial hiddens to augment and regularize, which is great!) but after your abstract and intro, I was expecting to see the Great Software 2.0 debugger, and this it is not.

The Self Composition section was exciting to read. It seems you get a big boost in uncertainty estimation using the "random walks through the reconstruction layers" trick, which is awesome. However, given the great cost of this self-composition (not to mention the cost of the decoder) it would be very useful to show in Table 1 what the relative costs in terms of computation time and model weights go along with the DecNN's and the various baselines. I imagine your approaches still win over Ensembles on both counts, but it would be useful to see the details to contextualize the cost of your idea.

The *Out-of-Distribution Detection* section was informative, but I was having trouble distilling how much of its results are due to the improved uncertainty estimation of RecDecNN and how much of it due to other reasons inherent in the DecNN architecture. Unless I misunderstand the experiments in this section, any black-box uncertainty estimator that gave improved results, would also improve OOD detection, regardless of the presence of input reconstruction, or the decoder sampling via recursion paths. In other words, although I buy this as an outcome of your work improving uncertainty estimation, I see it as a secondary benefit that's not a primary contribution of your work. Please clarify if I've misunderstood your contribution here. The same applies to Calibration.

I have a somewhat similar concern about section 5.2 and your composition with pre-trained models. It seems that this composability argument is, again, rather superficial. Perhaps a way to argue that DecNNs are better suited to this would be to show that if you move the $m$ model into $f$, you get inferior results. Perhaps another way to ask this is how much would standard models have benefited from the pre-trained regularizer (if your new term in equation 7 didn't involve $g_\phi$, but just $(f_\theta(x) | m(x)))? Is there something inherent to DecNNs that makes this regularization better?

One question about your fairness section is how the experiments shown in Table 6 affect primary performance (overall accuracy in Table1). I'd love to see that in a future version of this work, although I find the results here very welcome.


# Smaller comments

* Lines 87--92. This is a nit really, but I find the notation for $h$ confusing, given that there's lots of $h$'s we're dealing with in most of the paper. I think it would be much cleaner to just change the expression $x \in S(h)$, into something like $x \in S(f_\theta(x))$? It changes nothing in your point here, and it doesn't add more cognitive muddiness.

* Line 256 is confusing. It sounds like you're incorporating a linear model to a residual network. Please rephrase.

* You use both ((Re)DecDNN and (Re)DecNN) in the paper. Please use one (I recommend the latter).

* Your Table 1 has no caption. Oops.


**Time Spent Reviewing:**

8

---

> ### Author Response · Authors · 2021-08-09
> **Author Comments to Reviewer gKMX**
>
> We thank the reviewer for their thoughtful comments. We hope to address questions and concerns below:
>
> > It would be good to describe the differences between DecNN and vanilla autoencoders in the main text.
>
> Good question! The primary differences are (1) there is a reconstruction error for every hidden layer in the network, not just the final layer; (2) DecNNs have a supervised objective, which critically changes what information is captured in the hidden layers (and the resulting reconstructions); (3) the reconstruction in Autoencoders are not reused where as in DecNN we investigate using the reconstruction as input to secondary models.
>
> > Is OOD detection and calibration a primary contribution of your work or a secondary artifact of improved uncertainty estimation?
>
> The main contribution of Recursive Self-Composition is improved uncertainty estimation. We measure the usefulness of uncertainty through OOD detection and calibration. It was not immediately obvious to us that uncertainty captured in our method would result in better calibrated models or be able to do OOD detection, corruption detection, or adversarial example detection. The results showing it could were thus interesting to us. Certainly, there are methods specialized for OOD detection or calibration that would likely outperform ReDecNN or MCDropout, but these methods generally require additional training / computation.
>
> > How does composition with pre-trained models compare to direct pre-trained regularization?
>
> This is a great question. We ran new experiments comparing Eq. 7 and direction regularization.
>
> Comparison of Objectives Composing Pretrained Linear Models (MNIST)
>
> Method | None | Equation 7 | Direct Regularization
>
> Accuracy | 97.50 | 97.17 | 96.72
>
> Misclassification | 0.869 | 0.904 | 0.828
>
> Adversarial | 0.787 | 0.805 | 0.707
>
> OOD (Fashion) | 0.812 | 0.845 | 0.699
>
> OOD (CelebA) | 0.893 | 0.928 | 0.782
>
> Corruption (Mean) | 0.727 | 0.790 | 0.667
>
> Calibration (ECE) | 0.334 | 0.296 | 0.280
>
> Intuitively, the main difference is that we found from Sec. 4 that reconstructed images in deeper layers are not identical to the original image in meaningful ways (e.g. information is dropped or edited). So, $m(x)$ and $m(g(h))$ can be quite different since $x \neq g(h)$. In Eq. 7, we believe constraining the model to be similar to what a pretrained model thinks of the reconstruction (which is a dynamic value that varies with the current weights and depth) is a stronger learning signal than what the pretrained model thinks of the original input (which is a static value). In other words, regularizing all $f(g(h_l))$ to be like $m(x)$ at all layers $l$ is too strict of a constraint. Although doing this results in better calibration (see Table above), we find worse uncertainty -- resulting in lower OOD, corruption, adversarial, and misclassification detection accuracy. We agree with the reviewer that this is an interesting ablation and will include these results in the supplement.
>
> > What is the time cost of self-composition versus an ensemble and standard models?
>
> Please see the general response above! We provided experiments measuring run-time cost across the different datasets and models we studied.
>
> > How does the fairness results affect overall accuracy?
>
> Good question! We measured overall accuracy on the predicting ‘Attractiveness’ for models in Table 6. We find little difference in performance across the models despite the notable differences in using protected attributes.
>
> Model | F1 | Accuracy
>
> Standard | 0.784 | 74.83%
>
> ReDecNN (Eq. 4) | 0.784 | 76.45%
>
> ReDecNN (Eq. 8) | 0.779 | 76.30%
>
> > Writing comments: line 256 is confusing, typos in naming, and missing caption.
>
> Thank you for pointing these out! We will make the suggested changes for the final version.

---

> > ### Comment · Reviewer_gKMX · 2021-08-28
> > **Response to author comments**
> >
> > Thank you for the detailed comments, both on my review and the other reviews. I must say I am impressed with the deep and earnest approach you took to all comments, and the non-trivial work you put in to respond to reviews and ethics reviews.
> >
> > I especially appreciate your pivot to a different protected attribute, to respond to the various comments on your choice of "fairness" task, as well as your experiments on the alternative use of pre-trained models (in response to my review).
> >
> > Although I started out positive with your work, I'm even more so now. Excellent work!

---

> > > ### Author Response · Authors · 2021-08-30
> > > **Additional response**
> > >
> > > Thank you for the positive notes!
> > >
> > > We appreciate the time you and the other reviewers took to provide helpful comments and suggestions. We think the new experiments and findings have made the paper better and we are excited to incorporate them in the final version.

---

### Official Review · Reviewer_9qhL · 2021-07-30

**Rating:** 7
**Confidence:** 4

**Summary:**

Rather than only building the forward model P(y|X), this work propose to also learn a generative "inverse" model P(X|h) for any hidden layer h. This ability mirrors traditional software debugging, where upon a failure computation, one can "pull back" from the point of failure and detect at which point is the program behaving erroneously. This paper explores how to cleverly leverage this trained "sampled pullback" for various applications such as probing, uncertainty estimation, calibration, and fairness.


**Limitations And Societal Impact:**

it is sufficient

**Main Review:**

# significance
I believe this work is significant. It is using a very simple idea, that of training an inverse sampler in conjunction with the forward NN. This work use the inverse in a clever way: realizing that with the inverse one can arbitrarily "compose" different neural networks (of same type signature) by pulling back the first NN to a sample of prototypes in the input space then passing it to the second NN. This clever insight enabled a range of applications, from building an ensemble for uncertainty estimation and calibration, to information-flow style privacy.

# originality
I cannot comment on this as I am not an active practitioner in explainable neural networks. However I have not seen approaches like this from my experience, so if anything this approach (using inverse as a mean of composition) is novel to me. I will defer this to other reviewers who are more fluent in the field.

# training method
this work is simple technically, train a conditional-vae-ish inverse sampler, and section 2 and 3 read very clear.

# applications
this paper claims that the simple training of inverse would allow application of probing, composition, and constraints.
below are some detailed comments

## probing
this section is clear, by looking at the inverted X from different layer, a user can reason about where in the network did something important such as "wearing hat" got lost.
I feel some results of probing can be relegated to the appendix, as probing is a mere consequence of training the inverse, and it is not very novel / exciting compared to composition. probing can be in fact a sub-section of section 3, entirely written in 4 sentences and the reader won't miss it.

## composition
overall this is the coolest, and most insightful part of the paper (section 5), it should have a higher prominence. It is currently buried very deep.

"But given two image classifiers, it is not clear how to do so" - why do I even want to do this? it'll be good to have a concrete use-case to motivate this upfront.
given f1 and f2, what is the interpretation of the "semantics" of f2(f1_inv(f1(x)) exactly? If the inverse is bijective, the semantics is simply f2(x), but f1 loses information along the way, so what is the intuition here, and can you write about it?
the prose """in other words ... joint objective""" reads a bit dense, I was able to understand it after a few reads, but it would benefit from a figure, as it is central to understanding this work.

The application of using self-composition to get an ensemble is good, and the explanation is very clear (I especially enjoyed the part on sampling from a space of O(L^D), very cool).

I think parts of section 5 is the actual main thesis of this paper, instead of this "building software 1.5" story which feels out of place.

to me, this paper is asking a fairly philosophical question: "What does it mean to compose two arbitrary functions which has no business of being composed, by sampling the inverse of the first one, f2(f1_inv(f1(x))?" The answer to this (abstract) question then opened the door for a host of insights on how to build neural network compositions that solve a range of tasks. it is unfortunate that, such an interesting insight is buried so deep in section 5, and the out of place "building software 1.5" would have prominence over this in the introduction.

anyone working on explainable NN can claim they are working on "software 1.5", but not anyone has the unique insight of composing functions by leveraging a sampled inverse, and using it to build an ensemble.

## constraining
this section is clear, the training objective function is very clever.
the analogy to assertions is not good, more details in clarity section.

## other modalities
sure, this is obvious.

# clarity
this is where I'd like to see some major improvements

### figure 1 looks like a hastily cropped figure 3
this figure should carry way more information, as this is the first thing the reader will draw attention to, yet it is the worst figure of them all.
currently, it is 1)vague caption, 2)has no annotation, 3)and the axis are not labeled, and 4)tells an incomplete story
1) what is misclassification mean? only after reading the whole paper did I realize that, the bottom row is an input that should be a "6" that is being missclassified as a "4".
2) the first 3 rows need caption as well, was the mistake "hair - nohair" for the first row? was it "man - woman" for the second row? was it "bag - dress" for the third row?
3) it is also unclear that the horizontal axis represents how "deep" into the network the latent vector / activation was when used to invert (figure 3 does a better job).
4) why are composition and constraints not highlighted equally here, why is only "probing" being highlighted? you have the phrase "improving compositionally" in the title, presumably this is the important story to tell, and should be reflected in figure1 somehow.

incidentally, the "probing" capability is the _least_ interesting, most obvious part of this paper, at least to me.

I am inclined to raise my score if the authors can provide an anonymous url to an updated figure_1.png that clearly explains what decNN is on a high level, how does it allow for arbitrary NN composition, and its applications (essentially, sets up the main technical contribution and "promise" of this work). maybe consult a colleague who is proficient at effective visual communication.

### the analogy to software 1.0 feels strained

For instance,
"""In traditional software, assertion statements are a popular way to make sure the program is executing
as expected. Ideally, we would have the same functionality in neural networks: the ability to specify
what information an activation should not capture. """
this analogy doesn't quite work, as "executing as expected" =/= "specify what information is kosher"
a better analogy is 'information flow", take a look : https://www.cs.cornell.edu/andru/papers/jsac/sm-jsac03.pdf

In general the analogies to software 1.0 feel strained (here, and in the introduction). My (personal) opinion is that the story of composition you have in this paper is already good enough, there really is no point trying to shoe-horn a story with software1.0, 2.0, 1.5 at all, which "get in the way" of the main thesis of this paper, which is composition of two functions of the same input type and the consequence of this composition.

### minor phrasings
"to expect “useless” information to be dropped deeper into the network" could change the word "drop" to "discard" as drop can also mean "to place" (drop me off at the zoo) which is confusing.

# overall
overall this is a solid paper with a simple approach that is easily applicable (at least in theory, provided that the inverse sampler is easy to train) to a range of important problems. the experimental results look solid to me as well, I especially liked the section on using self composition as a way of building an ensemble, which in turn can be used for uncertainty estimation and calibration.

it is being held back by poor organization and story telling (perhaps a lack of re-writes on the paper) and a negligence of effective visual communications, which hampers its readability in its current form. this is unfortunate, because this work is very inspiring to me, both on a conceptual level (composing two arbitrary functions that takes the same input type using inverse) and on a practical level (the preliminary results looked promising). Practically, I think this work would benefit by moving some contents into the appendix to free up more space for better story telling.

**Time Spent Reviewing:**

4

---

> ### Author Response · Authors · 2021-08-09
> **Author Comments to Reviewer 9qhL**
>
> We thank the reviewer for their thoughtful comments. We hope to address questions and concerns below:
>
> > “I am inclined to raise my score if the authors can provide an anonymous url to an updated figure_1.png that clearly explains what DecNN is on a high level.”
>
> We agree that Figure 1 can be vastly improved. We have made a new version that focuses less on probing but describes DecNN and its applications here: https://drive.google.com/file/d/1Qmn-4T2NZlCHANjSJ4fk6BttI-JScExd/view. (This Figure will also replace the current Figure 2 on page 2). We hope this figure is more thorough and clear and will include it in the final version if reviewers prefer.
>
> > Probing is a less exciting application than compositionality and constraints, and this should be reflected in the writing.
>
> We agree that Section 5 and 6 are the more novel applications. In the final version, we will greatly shorten Section 4 and make it a subsection instead, moving most of the details to the appendix. This way, the method description will be immediately followed by Section 5 & 6.
>
> > The current explanation in lines 149-152 is too dense and could use a figure.
>
> We will improve the phrasing here. To clarify, because the input $x$ and the decoded activation $g(h)$ are of the same type (where $g$ is a generative model ‘inverting’ the function $f_1$ and $h = f_1(x)$), we can reuse $g(h)$ as input to a function $f_2$ that would normally take $x$ as input. Note that if $f_2 = f_1$, we call this “self-composition”. As suggested, we will add a new figure to the supplement: https://drive.google.com/file/d/1RKDNDxVDKBMuRjTp_Oza2nefCZ94Pr57/view.
>
> > Definition of misclassification is unclear.
>
> We will be sure to explicitly define misclassification before usage. To clarify, we are measuring differences in model uncertainty when the model correctly classifies an example versus when the highest predicted probability class does not match the ground truth label.
>
> > "But given two image classifiers, it is not clear how to do so" Why do I even want to do this? If $f_1$ loses information along the way, what is the intuition here?
>
> This is a good question! One motivation we drew upon was to use one model $f_2$ as a way to interpret and/or edit what another model $f_1$ is learning (although $f_1$ loses information, $f_2$’s role is to gauge and control what information is lost). In the paper, we explored (1) regularizing a neural network $f_1$ using a simpler model $f_2$; (2) encouraging $f_1$ to ignore a protected feature with $f_2$ as a surrogate measure. We can also view self-composition (where $f_1$ = $f_2$) as a form of regularization via data augmentation. We will add this idea as motivation when introducing the method in the main text. In our opinion, there are also future applications to explore reusing what $f_1$ has learned to more efficiently train $f_2$. For instance, reusing information in the hidden states learned for a segmentation model to train a model more efficiently for detection (since the two tasks are very related).
>
> > Phrasing fixes: “drop” -> “discard”; better prose for “in other words … joint objective”.
>
> Thank you for catching these! We will correct them and smooth the writing.

---

> > ### Comment · Reviewer_9qhL · 2021-08-13
> > **thanks for the comments**
> >
> > I agree with the gKMX 's comment on "this is a bit over selling" in the introduction, this is aligned with my sentiment of "the analogy feels strained". I think this paper is sufficiently interesting technically, there is no need to make extra analogies other than "you want composition? here is a very novel way of doing it which allows you to prove all different layers, without having to re-train and adapt any weights".
> >
> > The technical section (once you got to it) reads super smooth, and the introduction / framing is the only thing that's holding this work back in my opinion. I'd try to incorporate what gKMX has pointed out to make this work stronger. I like this work and I'd want to see it get the right framing / exposure that works in a tight way without being too broad / cringe in its initial claims.
> >
> > Updated figure 1 looks good to me. Raising my score to 7.

---

> > > ### Author Response · Authors · 2021-08-16
> > > **Thanks for the additional feedback**
> > >
> > > We agree with both reviewers 9qhL and gKMX that the introduction can be rewritten to be more direct. In the final draft, we will take both reviewer's comments and tighten the text to more directly motivate the method from a composition / architecture angle rather than relying too much on the software 2.0 analogy.

---

> > ### Comment · Reviewer_gKMX · 2021-08-28
> > **Drive-by thanks for the comments**
> >
> > I didn't ask the questions you're responding to, but I definitely appreciated the answers. The new figures look very helpful and make a much better case (in my opinion) about composition than the submitted version of the paper did.

---

### Review · Ethics_Reviewer_HrxU · 2021-08-12

**Recommendation:**

Remove the discussion about fairness. The paper doesn’t need it and it gives the discussion a sheen of social responsibility that isn’t warranted.

**Ethics Review:**

One reviewer is concerned about the use of an “attractiveness” task to illustrate the method. I don’t think this is an issue, but I do question the claim that this can be used as part of a fair classifier. Such a claim needs a concrete domain, a clear understanding of a context, and better comparisons. I would have been happier with a different example to illustrate the idea of ignoring an attribute without implying that this helps with fair decision making, which is unclear at best.

---

> ### Author Response · Authors · 2021-08-15
> **Author response to ethics reviewer HrxU**
>
> We agree that the method presents a way to encourage classifiers to ignore attributes, which is a separate idea from fairness. We also acknowledge that much more research is needed to truly claim fairness as a consequence of the method. So, for the final draft, as the reviewer suggested, we will remove the discussion around fairness and narrow the claim just to imposing constraints on the learned decision function, which we find interesting in it of itself.

---

### Review · Ethics_Reviewer_mu5w · 2021-08-12

**Recommendation:**

I would recommend that the authors 1) dedicate a brief section to describing the data they are using and tasks that they are attempting to predict, 2) either write a statement on the ethical issues of CelebA or replace or drop the experiments using the dataset. There is no clear information or reasoning as to why the tasks chosen have been chosen, however the method, if my reading is correct, should be invariant to the task so I see no reason why the authors couldn't or shouldn't replace the task with another. Moreover, by having a task and dataset that has a greater domain shift from the other datasets used and tasks explored would more clearly show the benefits of the method across domains, and thus strengthen the contributions of the paper.

**Ethical Issues:**

Yes

**Ethics Review:**

This manuscript works on the CelebA dataset on the task of predicting attractiveness. As a task, this is pretty non-sensical as attractiveness is in the eye of the beholder and subject to quite strict cultural norms of beauty. Thus, predicting attractiveness as annotated reveals more about the annotators than attractiveness. Given global euro-centric standards of beauty, we can safely assume that this task reduces to predicting centrality to predominately white European features - this is the first issue. The second issue is a writing issue, information on the task that the authors are conducting is buried in the paper and it must be read quite closely to identify what tasks the authors are trying predict. The prediction tasks that the authors perform are auxiliary to the method, so it is also not clear why the authors choose the specific tasks. This information should be readily available in the manuscript.

Please see more detailed comments below.

“prediction task” (“Improving Compositionality of Neural Networks by Decoding Representations to Inputs”, p. 5) What the authors are predicting one predicting attractiveness is not in fact attractiveness but instead association with whiteness and Euro-centric standards of beauty. The rationale for selecting this task for showing the utility of the method is unclear to me. Given that the question of whether someone is attractive or not is not a question devoid of politics, I would recommend that the authors simply remove the experiments predicting attractiveness or replace them with something that is less overtly fraught with issues.Using the example of predicting attractiveness opens up this paper to a lot of criticism that has nothing to do with the method or what the paper and the authors set forth to do and it’s for this reason that I recommend that the authors either drop the experiments or in predicting attractiveness or replace them with something else.Additionally, I would recommend that the authors more clearly state what tasks they are setting out to perform. Reading exactly what the authors are trying to do on the different data sites is unclear unless you are familiar with the dataset and the standard tasks on them and, moreover it is always good practice to describe your data and your classification tasks

---

> ### Author Response · Authors · 2021-08-15
> **Author response to ethics reviewer mu5w**
>
> We thank the reviewer for their thoughtful analysis of an important issue. We find the points raised to be beneficial to the paper and will address both.
>
> To the reviewer’s first point, we agree that predicting attractiveness is an ill-posed task with inherent bias built into the annotation. We also agree with the reviewer that research should stray away from propagating this task forward. As the reviewer aptly noted, the claim of our proposed method is to constrain what information a classifier learns, which is not bound to the attractiveness task. We are working to change the task. In our opinion, constraining computation (and Equation 8) is an interesting application of decodable neural networks and we prefer to not remove the section completely. However, we can easily change to more sensible attributes in the CelebA collection with minimal bias in the annotation process (e.g. wearing a hat vs. having hair). We are currently running a few experiments for this now. If appropriate and successful, we will include these new results in the final draft in lieu of Table 6. We will also post the new results to the forum once they complete in the next few days.
>
> To the reviewer’s second point, we agree that we can be more clear on what the tasks in the experiments exactly are. In the final draft, we will (1) add a datasets section in the supplement summarizing all the datasets and tasks with a reference in the main text and (2) before each set of experiments (current lines 126-129, 193-195, 206-210, 220-224, 240-247, 256-262, and 277-285), we will add a description of the inputs and outputs being used, especially in cases where we are using a dataset in a non-standard way (e.g. for OOD, or for ignoring an attribute).

---

> > ### Author Response · Authors · 2021-08-20
> > **Initial results from new experiments**
> >
> > As mentioned in the comment above, we ran a new experiment replacing the ill-posed attractiveness task. Here, we again consider two attributes from CelebA: "bald vs not bald", and "bearded vs not bearded". We chose these attributes as they are much less subjective than attractiveness and thus hopefully less susceptible to obvious annotation bias. We compare training a standard neural network for predicting bearded-ness to an unconstrained ReDecNN (Eq. 4) to a constrained ReDecNN (Eq. 8). Like in the original Table 6, we compare F1, AP, and ECE between two groups: a test set of only bald individuals and a test set of not-bald individuals.
> >
> > Model   F1 (Bald / Not Bald)    AP (Bald / Not Bald)    ECE (Bald / Not Bald)
> >
> > Standard    0.338 / 0.438 (0.10)   19.4 / 23.9 (4.5)   4.76 / 3.56 (1.20)
> >
> > ReDecNN (Eq. 4) 0.327 / 0.484 (0.15)   18.6 / 22.3 (3.6)   4.74 / 3.54 (1.20)
> >
> > ReDecNN (Eq. 8) 0.461 / 0.501 (0.03)   27.2 / 28.9 (1.6)   3.80 / 3.28 (0.52)
> >
> > In the table above, the number in parentheses is the difference between the two groups. Note the F1 and AP for this task are generally lower than the attractiveness task in Table 6 as the labels of bearded vs non-bearded individuals are less balanced. Still, we observe that by constraining computation with Eq. 8, we find more balanced F1, AP, and ECE across the two groups than with either baseline. Taking a step back, this experiment represents a possible alternative to Table 6. We will investigate a few more attributes for the final draft.

---

### Decision · Program_Chairs · 2021-09-28

**Decision:**

Accept (Poster)

**Comment:**

UPDATE: The revision has been reviewed and the paper is officially accepted.

----

After extensive discussions among SACs, ACs, and the program chairs, we have decided to conditionally accept this paper. The primary concerns were around over-claiming, the interpretation of results, and the analogy to programming (which was viewed as a distraction from the main contributions of the paper).  There was some debate around whether these issues constitute a "fatal flaw" that should prevent the paper from being published at NeurIPS, but all agreed that these issues need to be addressed. We hope that the authors will take into account feedback from the reviewers and make the changes that were promised as part of the rebuttal.

Additionally, in order to be accepted, the following changes must be made in the revision:

Introduction:

L52: "This joint training optimizes the classifier not only to be highly performant, but constraints its activations to a manifold that the generative model is able to decode. Consequently, the networks entire computation is constrained to a 'decodable' vector space... " This is misleading and should be rephrased or removed; just because the activations at a given layer can decode the input doesn't mean that that information is being used by the downstream layers to actually make decisions.

L59: This paragraph should focus on what the paper actually demonstrates empirically; essentially remove the first sentence.

Section 3:

Highlight that equation (4) doesn't require that the reconstructed images $g_\phi(h_l)$ actually produce the same output y, so the reconstruction may not actually be highlighting the features that were important for f to make its decision in the first place.

Section 4:

This section should be significantly rewritten to avoid making claims unsupported by the evidence. Specifically:

L116: "In practice, this theory leads us to expect 'useless' information..." This would be true if Equation (4) wasn't rewarding the network for keeping the useless information to help support the decoding task.

L119: This paragraph is misleading and should be removed. While it is true that you can visualize the information at each layer, you cannot actually tell which of that information is actually being relied upon to make the decision.

L130: This is the only paragraph in this section that states an empirical fact; the whole section should focus more on what this experiment objectively shows, but avoid making claims not supported by the observations made in this experiment.

L138: This paragraph again misrepresents what can really be inferred from the generated images.

Section 6.

The analogy with assertions is misleading as it implies a level of guarantees that is not supported by the evidence. The section should focus more closely on what the experiments actually demonstrate for this application. The attractiveness task should also be replaced with the "bald vs not bald", and "bearded vs not bearded" experiment presented in the rebuttal.

Abstract:

The abstract should be consistent with all the changes made to the rest of the paper.

----

The original meta-review for this copy of the paper follows:

I am really torn about this paper. On the one hand, I think this paper presents an intriguing and original idea: train a generative model to reconstruct the input from the intermediate layers of the network, and then feed those reconstructed inputs either to the same network or to other networks that perform related tasks, possibly even recursively. The paper then presents empirical evidence that these compositions can be engineered to have some useful properties, such as identifying when their input is out of distribution, or avoiding the use of certain types of information in a classification.

On the other hand, the paper as written suffers from extreme over-claiming and interprets many of its results in a way that I don't think is warranted by the data. The gap between the clear contributions of the paper and the narrative that the authors built around them was pointed out by multiple reviewers. While the authors did significant work through the rebuttal period to address many of the reviewer concerns and made some significant promises in how they were going to rewrite the paper. I think the issues are significant enough that I really think a fresh version of this paper that incorporates all the proposed changes needs to be reviewed by a fresh panel. The issues with his paper are too big to accept it without further review.

I now describe some of the issues that I found most questionable about this paper.

The paper is organized around an analogy to traditional programming and program debugging. The claim is that the generative models allow us to interpret what each of the layers in the network mean so that they can be debugged probed, asserted, etc. However, the analogy doesn't really stand up to scrutiny, and in arguing for this interpretation of its results, the paper fails to explore alternative explanations for the observed behaviors.

For example, consider section 4. The objective empirical observation is that for this set of images and image classification tasks, when an image is misclassified, the reconstructed images progressively look more and more like the incorrect label. This is interesting, but the authors try to claim something much bigger; they claim that these images can allow one to understand "what features a neural network is attending to in making a prediction, or to understand what features make it susceptible to making an error". I find this highly unlikely. In fact, the loss function in Equation 4 almost guarantees that this would not be the case. For example, suppose that in the case of the handbag, the neural network actually learns that a handful of pixels making up the handle of a handbag are what characterizes it as a handbag. An image that allows one to understand what features a neural network is attending would show just the handle of the handbag and discard everything else if that's what the network is using for its classification. However, equation 4 rewards the network for maintaining enough information at each layer to reconstruct the whole handbag, even if most of that information is not actually being used to make the prediction, only for the reconstruction. In fact, the handle of the handbag has very few pixels, so the reconstruction network would not be significantly penalized if it failed to display the handle; there is nothing in equation 4 that actually requires the reconstruction to focus on the features that are important for classification.

There are similar issues with each of the subsequent sections. They are motivated and explained in terms of this analogy to programming, but the analogy just doesn't hold up to scrutiny. I think section 6 is particularly problematic because the analogy with assertions implies a level of guarantees that is simply not supported by the evidence. Now, I actually think this section is the most interesting one in the paper, and I think it is unfortunate that the authors chose such a bad example to make their point. However, I think a lot more evidence is needed to show that this approach can be used to reliably prevent the network from using a protected characteristic when making a decision (that would be a great paper by itself). For example, Equation 8 forces the generative model to optimize three criteria: faithfulness to the input image, randomness with respect to the protected characteristic, and ability to recreate the correct classification output. How these competing goals are balanced out will depend on some hyperparameter tuning that may determine the extent to which the resulting network actually satisfies the desired constraint. For example, a protected characteristic that affects a lot of pixels in the image will increase the cost of adding diversity relative to the goal of being faithful to the input image and may require different hyperparameters from one that only occupies a few pixels.

Overall I think there is an important and novel contribution in this paper, and I do believe the authors understand some of the issues with the paper as submitted and have demonstrated a willingness to rewrite the paper based on the criticisms in the reviews. This is reflected in the high scores given to the paper by all the reviewers. However, multiple reviewers raised the same concern highlighted above, and the Area Chair feels that this concern is quite substantial and requires a major revision to address.

**Consistency Experiment:**

NeurIPS has a long history of experimentation. In 2014, NeurIPS ran an experiment in which 10% of submissions were reviewed by two independent committees to quantify the randomness in the review process. This year, we repeated a variant of this experiment to see how the quality of the review process has changed over time.  This paper was part of the experiment and was therefore assigned to two committees (consisting of reviewers, an Area Chair, and a Senior Area Chair) that reached independent decisions.  If both committees made the same recommendation, this recommendation was followed. If a single committee recommended acceptance, the paper was accepted (with the exception of a few cases in which the other committee identified what we considered a fatal flaw, e.g., an error in a key result).

This copy’s committee reached the following decision: **Reject**

The other committee assigned to the paper recommended **Accept (Poster)**.  You can find the other set of reviews, along with any follow up discussion with the authors here:
https://openreview.net/forum?id=ms1fOdxXhWH